# Trajectories of healthy ageing among older adults with multimorbidity: A growth mixture model using harmonised data from eight ATHLOS cohorts

Hai Nguyen[1]*, Dario Moreno-Agostino[1], Kia-Chong Chua[2], Silia Vitoratou[3], A. Matthew Prina[1]

1 Health Service and Population Research Department, Institute of Psychiatry, Psychology and Neuroscience, King's College London, London, United Kingdom, 2 Centre for Implementation Science, Institute of Psychiatry, Psychology and Neuroscience, King's College London, London, United Kingdom, 3 Biostatistics and Health Informatics Department, Psychometrics and Measurement Lab, Institute of Psychiatry, Psychology and Neuroscience, King's College London, London, United Kingdom

* hai.nguyen@kcl.ac.uk

## Abstract

### Objectives

In this study we aimed to 1) describe healthy ageing trajectory patterns, 2) examine the association between multimorbidity and patterns of healthy ageing trajectories, and 3) evaluate how different groups of diseases might affect the projection of healthy ageing trajectories over time.

### Setting and participants

Our study was based on 130880 individuals from the Ageing Trajectories of Health: Longitudinal Opportunities and Synergies (ATHLOS) harmonised dataset, as well as 9171 individuals from Waves 2–7 of the English Longitudinal Study of Ageing (ELSA).

### Methods

Using a healthy ageing index score, which comprised 41 items, covering various domains of health and ageing, as outcome, we employed the growth mixture model approach to identify the latent classes of individuals with different healthy ageing trajectories. A multinomial logistic regression was conducted to assess if and how multimorbidity status and multimorbidity patterns were associated with changes in healthy ageing, controlled for sociodemographic and lifestyle risk factors.

### Results

Three similar patterns of healthy ageing trajectories were identified in the ATHLOS and ELSA datasets: 1) a 'high stable' group (76% in ATHLOS, 61% in ELSA), 2) a 'low stable' group (22% in ATHLOS, 36% in ELSA) and 3) a 'rapid decline' group (2% in ATHLOS, 3% in ELSA). Those with multimorbidity were 1.7 times (OR = 1.7, 95% CI: 1.4–2.1) more likely to

**Data Availability Statement:** All relevant data are within the manuscript and its Supporting Information files.

**Funding:** This work was supported by the European Union's Horizon 2020 Research and Innovation Programme [under grant agreement number 635316], as part of the ATHLOS Consortium (Ageing Trajectories of Health: Longitudinal Opportunities and Synergies, http://athlosproject.eu/). MP is partly funded by the MRC (MR/S028188/1, MR/T037423/1 and MR/T038500/1). SV was funded by the Biomedical Research Centre for Mental Health at South London and Maudsley NHS Foundation Trust and King's College London.

**Competing interests:** The authors have declared that no competing interests exist.

be in the 'rapid decline' group and 11.7 times (OR = 11.7 95% CI: 10.9–12.6) more likely to be in the 'low stable' group, compared with people without multimorbidity. The cardiorespiratory/arthritis/cataracts group was associated with both the 'rapid decline' and the 'low stable' groups (OR = 2.1, 95% CI: 1.2–3.8 and OR = 9.8, 95% CI: 7.5–12.7 respectively).

## Conclusion

Healthy ageing is heterogeneous. While multimorbidity was associated with higher odds of having poorer healthy ageing trajectories, the extent to which healthy ageing trajectories were projected to decline depended on the specific patterns of multimorbidity.

## Introduction

Over the last few decades, there has been a shift in the focus of ageing studies, from a disease-oriented approach to a health-oriented approach [1]. Ageing is recognised as a dynamic process [2] in which intrinsic capacity (i.e. physical and mental capacities), functional ability (i.e. health-related attributes that enables people to be and do what they value) and the environment interact [3]. The World Health Organisation (WHO), in their World Report on Ageing and Health, highlighted the concept of healthy ageing, which was defined as "the process of developing and maintaining the functional ability that enables well-being in older age" [3]. Central to this concept is the recognition that neither intrinsic capacity nor functional ability remains constant over time [3]. Both tend to decline, but the rate at which they decrease may be different, depending on people's life choices or interventions at any point during their life course [3]. To understand how people age and to respond to their healthcare needs, it is important to study how healthy ageing changes over time and what factors are associated with this process.

A new measure of healthy ageing, the healthy ageing index (HAI) score, has been recently developed [4]. As healthy ageing is a heterogeneous process, by analysing the growth patterns of the HAI score in the population, healthy ageing trajectories can be modelled. Few studies have attempted to investigate healthy ageing trajectory patterns in older adults [5–7]. Several patterns have been identified when examining the impact of lifestyle behaviours on healthy ageing [5,8]. However, there is little evidence about how these patterns might be different for people with multimorbidity. As with the ageing process, multimorbidity (i.e. the co-existence of two or more chronic diseases in the same individual) is heterogeneous [9]. This is demonstrated by the diversity of its operationalisation and patterns. Seventeen measures of multimorbidity and 63 patterns of three or more chronic diseases, many of which were unexplained or unspecified (e.g. none of the diseases were overrepresented nor share common pathological pathways) have been identified across studies [10,11]. There has been evidence that different clusters of diseases were associated with different health outcomes [12,13]. Certain disease groups appeared to be more likely to be associated with mortality [14], reduced quality of life [15], disability [16] and lower self-rated health [17] than others. While multimorbidity (both its presence/absence and patterns) at baseline was found to be negatively associated with baseline healthy ageing [13], it is not clear how multimorbidity relates to the course of healthy ageing over time.

Drawing on this evidence, we hypothesised that individuals with multimorbidity were more likely to have poorer healthy ageing trajectories than those who have none or one disease. However, among people with multimorbidity the degree to which their healthy ageing

trajectories were projected would be determined by the patterns of multimorbidity. Our objectives were therefore 1) to describe different patterns of healthy ageing trajectories that existed among older adults globally, 2) to examine if multimorbidity was associated with different patterns of healthy ageing trajectories, and 3) to evaluate how different multimorbidity patterns might be related to the projection of healthy ageing trajectories.

## Materials and methods

### Data and sample

In this study, we used data from the Ageing Trajectories of Health: Longitudinal Opportunities and Synergies (ATHLOS) project. The ATHLOS project is a study of healthy ageing, which sought to gain a better understanding of the impact of ageing on health, as well as identify patterns of healthy ageing trajectories and their determinants [18]. This was achieved through the development of a new single measure of health status (i.e. the healthy ageing index score) [18]. Data from ATHLOS comprised of 16 existing longitudinal studies in 38 countries [18]. Similar variables which were in different formats across 16 studies were transformed and merged together to create a harmonised dataset. The harmonisation of variables was based on an iterative process of consensus of experts from the ATHLOS Consortium and publicly available on an online repository service platform (https://github.com/athlosproject/athlos-project.github.io/) [8].

In the present study, we included only studies that had at least three waves to enable investigation into changes of healthy ageing trajectories. These studies were the Australian Longitudinal Study of Ageing (ALSA) [19], the English Longitudinal Study of Ageing (ELSA) [20], the Study on Cardiovascular Health, Nutrition and Frailty in Older Adults in Spain (ENRICA) [21], the Health and Retirement Study (HRS) [22], the Japanese Study on Ageing and Retirement (JSTAR) [23], the Korean Longitudinal Study on Health and Ageing (KLOSA) [24], the Mexican Health and Ageing Study (MHAS) [25] and the Survey of Health, Ageing and Retirement in Europe (SHARE) [26]. The ATHLOS analytic sample consisted of 130880 participants (55.5% female) with an average age of 62.8 years (SD: 10.1) at baseline.

While ATHLOS data covered a large sample, which enabled the longitudinal examination of healthy ageing trajectory patterns at a global level, the lack of information on some health conditions across different cohorts hindered investigations into how these patterns might change among people with specific combinations of diseases. We therefore carried out a further analysis at national level using data from Waves 2–7 of the English Longitudinal Study of Ageing (ELSA) to explore how multimorbidity patterns might affect long term healthy ageing trajectories. ELSA, which is also one of the ATHLOS datasets, commenced in 2002 and participants at baseline were followed up every two years [20]. Instead of ELSA Wave 1, we used ELSA Wave 2 (2004/2005) as baseline, since in this wave extra biomarker data such as cholesterol level, blood glucose level, blood pressure, lung capacity were collected during nurse visits. These allowed us to derive more health conditions, such as chronic obstructive pulmonary disease (COPD), anaemia, blood clotting disorder, dyslipidaemia and obesity. At Wave 2, the ELSA sample consisted of 9171 participants (55.5% female) with an average age of 66.4 years (SD: 10.3). These participants were followed up until Wave 7 and their risk factors at baseline were used to predict their healthy ageing trajectories over a period of 12 years (six waves).

### Outcome variable

We measured patterns of healthy ageing trajectories using the healthy ageing index (HAI) score, a novel health metric developed by ATHLOS researchers [8,27–29]. The HAI score was computed at individual level across waves using Item Response Theory (IRT) framework [30].

IRT describes the relationship between an individual's 'trait' (unobserved characteristic or attribute) and how they respond to items on a scale [31]. In this study, a two-parameter logistic IRT model was implemented. Specifically, two parameters were estimated for each item: difficulty (reflecting the level of the latent trait where the probability of endorsing a particular item was 0.50) and discrimination (reflecting the items ability to discriminate between different levels of the latent trait around the difficulty level). Thus, the items contributed differently when estimating the respondents' levels in the latent trait [4]. The two-parameter logistic IRT model estimated the latent trait score, which comprised 41 items (both self-reported and measured by performance tests), covering various domains of health and ageing (S1 Table). These included vitality, sensory skills, mobility, cognition psychological symptoms and activities and instrumental activities of daily living (ADLs and IADLs) [18]. The adequacy of the IRT model as a measurement scale was tested using the Root Mean Square Error of Approximation (RMSEA; good fit <0.06), the Comparative Fit Index (CFI; good fit >0.95) and the Tucker-Lewis index (TLI; good fit >0.95) (4). The results showed that IRT model converged successfully with an excellent fit (RMSEA = 0.03, TLI = 0.99 and CFI = 0.99) and had a marginal reliability of 0.83 [8]. The score of each individual was calculated using the expected a posteriori estimation method [4]. To avoid negative values, the estimated latent trait score was transformed from Z-scores to T-scores based on a mean of 50 and standard deviation of 10. The construction of the HAI score has been described in detail elsewhere [4]. Evidence on the validity of the HAI score has also been provided in previous studies [27–29]. The lower the score, the less healthy a person was considered to be [8,27–29]. Since our aims were to identify the patterns of healthy ageing trajectories and how these patterns were associated with multimorbidity, only participants with the HAI score for at least two waves were included. Baseline characteristics of the analytic sample and the excluded sample are summarised in S2 Table.

## Independent variables

**Multimorbidity status.** Multimorbidity status was used as a key predictor of healthy ageing trajectories patterns in the ATHLOS dataset. To determine whether an individual had multimorbidity, we adopted the disease count approach, for this was the most common measure of multimorbidity. Eight diseases were included in the ATHLOS dataset, namely, diabetes, asthma, chronic pulmonary disease, hypertension, arthritis, angina, myocardial infarction and stroke. These eight diseases were common across 16 studies included in the ATHLOS dataset, and were self-reported by the participants, following the question "Has a doctor ever told you that you have/have had any of the following disease?". Each disease was coded as a dichotomous variable where presence = 1 and absence = 0. A multimorbidity score was derived, which was the sum of all the diseases. If the multimorbidity score ≥ 2, an individual was deemed to have multimorbidity (multimorbidity status = 1). If the multimorbidity score < 2, the individual was deemed not to have multimorbidity (multimorbidity status = 0).

**Multimorbidity patterns.** Multimorbidity patterns were used as a key predictor of healthy ageing trajectories patterns in the ELSA dataset. They were determined from 26 diseases at ELSA Wave 2 (both self-reported and objectively measured during nurse visits). These diseases included hypertension, angina, heart murmur, myocardial infarction, heart arrythmia, diabetes, stroke, asthma, chronic lung disease, COPD, arthritis, osteoporosis, cancer, diabetic eye disease, cataracts, glaucoma, macular degeneration, dementia, obesity, anaemia and iron deficiency, blood clotting disorder, psychiatric disorder, hyperlipidaemia, hypertriglyceridemia, hypoalphalipoproteinemia and high triglyceride/high density lipoprotein ratio. From these 26 diseases, three patterns of multimorbidity were identified using the latent class analysis (LCA) method: 1) the cardiorespiratory/arthritis/cataracts group (20% of the ELSA Wave 2

sample, with the highest prevalence in myocardial infarction, arrythmia, COPD, arthritis and cataracts), 2) the metabolic group (9%, with the highest prevalence in diabetes, hypertriglyceridemia, hypoalphalipoproteinemia and high triglyceride/HDL ratio) and 3) the relatively healthy group (71%, with the lowest prevalence in most diseases) [32]. Details of the coding and selection of diseases, as well as the description of the LCA method were described in our previous study [32].

**Sociodemographic and lifestyle covariates.** In addition to multimorbidity status (and multimorbidity patterns in the case of the models performed with the ELSA dataset), we also adjusted for sociodemographic factors (i.e. age, sex, education, net financial wealth) and lifestyle behaviours (i.e. smoking, drinking and physical activity level). Education was divided into three categories: 'less than primary or primary', 'secondary' and 'tertiary'. Drinking was measured by how often an individual had an alcoholic drink during the last 12 months and was grouped into three categories: 'never', 'rare' and 'often'. Household wealth was measured as within-country quintiles (with quintile 1 being the lowest wealth and quintile 5 the highest). Smoking status was defined as 'never smoked' and 'ever smoked' and physical activity was classified as 'sedentary or low', 'moderate' and 'high'.

## Statistical analysis

To explore differences in the ageing process among the study participants, we employed a growth mixture modelling (GMM) analysis. GMM allowed within-person growth trajectories and between-person variations in the trajectories to be modelled [33,34]. It accommodated population heterogeneity and enabled unobserved subgroups of participants with distinct patterns of healthy ageing trajectories to be identified [33,34]. GMM was performed separately for ATHLOS and ELSA data. In ATHLOS, GMM projected healthy ageing trajectories over 11 time points (covering a period of up to 22 years) while in ELSA, GMM was used to estimate the changes in healthy ageing trajectories over six time points (covering a period of up to 10 years). The average elapsed time across waves in both ATHLOS and ELSA was two years. Since the follow-up period of each of the ATHLOS cohort was different, a sensitivity analysis was carried out using ATHLOS Waves 1–3 where there was representation of all eight cohorts.

We first obtained the baseline growth model to find the best representation of change in healthy ageing trajectories [33]. Since health was often observed to decline with age, the change in healthy ageing trajectories was modelled as a linear change [12]. The optimal number of latent trajectory classes was determined by assessing the Akaike, Bayesian, sample-size adjusted Bayesian Information Criteria (AIC, BIC and SABIC respectively) [35–37], entropy statistic [38], the Vuong-Lo-Mendell-Rubin likelihood ratio test (LMR LR) [39], the Lo-Mendell-Rubin adjusted likelihood ratio test (aLMR LR) [39], the parametric bootstrapped likelihood ratio test (BLRT) [40], class size and average posterior probabilities for each latent class. The best fitting model with the optimal number of latent trajectory classes satisfied the following criteria: 1) lowest AIC, BIC and SABIC values, 2) p-value < 0.001 for LMR LR, aLMR LR and BLRT, 3) highest entropy statistic, 4) no class size < 1% of the whole sample and 5) posterior probabilities for each latent class ≥0.70 [38]. To balance model flexibility and estimation stability, the residual variances, which indicated within-class heterogeneity, were constrained to be equal across latent classes [41]. This approach was chosen over the alternative specification, where residual variances were constrained to be zero, because it reflected the individual heterogeneity within each latent class [41]. Missing values were handled by full information maximum likelihood technique, assuming missing-at-random (MAR).

Once the optimal number of latent trajectory classes was determined, we used a three-step approach introduced by Asparouhov & Muthen [42] to estimate the association of

multimorbidity (status and patterns) and sociodemographic/lifestyle covariates with each latent class. In this approach, the latent class model was estimated in step 1 [42]. Then in step 2, the most likely latent class variable was created for each individual using the latent class posterior distribution obtained in step 1 [42]. Finally, in step 3, a new model was estimated to evaluate the impact of predictor variables on the class membership, with measurement errors fixed to values obtained in step 2 [8,42]. The potential source of confounding posed by the heterogeneity of different cohorts included in the analysis was also accounted for by including the study variable in this step. The three-step approach was employed in our study since it accounted for membership misclassification and reduced errors due to posterior probability-based assignments [42]. In step 3, the largest size class was chosen as the reference. Where participants' sociodemographic and lifestyle behaviour details were not reported at baseline but were available in subsequent waves, these data were used in place of baseline data. Listwise deletion was therefore applied to observations with missing covariate values at all waves. GMM was conducted using MPlus version 8.2 [43].

## Results

### ATHLOS sample characteristics

The average HAI score in the ATHLOS sample fluctuated slightly over 11 time points: 51.0 (SD: 8.9) at baseline, 50.9 (SD: 9.3) at Wave 2, 48.9 (SD: 9.9) at Wave 3, 50.7 (SD: 9.6) at Wave 4, 50.4 (SD: 10.0) at Wave 5, 48.7 (SD: 10.1) at Wave 6, 49.0 (SD: 10.2) at Wave 7, 48.2 (SD: 10.6) at Wave 8, 48.2 (SD: 10.5) at Wave 9, 48.3 (SD: 10.7) at Wave 10 and 48.1 (SD: 10.7) at Wave 11. One quarter of the ATHLOS sample (25.1%) was classified as having multimorbidity at baseline. Table 1 provides the baseline characteristics of each of the studies that made up the ATHLOS sample.

### Multimorbidity status and healthy ageing trajectory patterns in ATHLOS

Results from step 1 of the three-step GMM procedure for ATHLOS, which considered four models with the number of classes ranging from two to five, is presented in Table 2. Based on the model fit information, the largest drop in AIC, BIC and SABIC was observed when the number of latent classes was increased from two to three. The likelihood ratio tests rejected the two-class model in favour of a model with at least three latent classes (p<0.001). Although the information criteria indices supported the four- and five-class models, further assessment showed that more than one class in these models had an average posterior probability lower than the threshold of 0.7 and comprised only 1% of the sample. Of the four models, the entropy statistic was highest for the three-class model (0.7), suggesting the classes were relatively well separated and the membership classification error was small. The three-class model was therefore selected as the optimal solution.

The patterns of healthy ageing trajectories that resulted from the selected model are shown in Fig 1. These included: 1) a 'high stable' group, which displayed a high level of healthy ageing at baseline and a slow decline over time (76% of the sample), 2) a 'low stable' group, which showed a low level of healthy ageing at baseline and a slow decline over the follow-up period (22%), and 3) a 'rapid decline' group, which presented a high level of healthy ageing at baseline but a steep downward slope over 11 waves (2%). Results from the sensitivity analysis using data from the first three waves of ATHLOS showed three identical patterns of healthy ageing trajectories (see S3 Table).

Table 3 presents results from the latent multinomial logistic regression analysis, with the 'high stable' group as reference, adjusted for sociodemographic and lifestyle covariates. The presence of multimorbidity significantly increased the likelihood of an individual being in the

**Table 1. ATHLOS baseline sample characteristics.**

| | ALSA N = 1851 (1.4%) | ELSA N = 14498 (11.1%) | ENRICA N = 2516 (1.9%) | HRS N = 32988 (25.2%) | JSTAR N = 3695 (2.8%) | KLOSA N = 8928 (6.8%) | MHAS N = 12925 (9.9%) | SHARE N = 53479 (40.9%) | ATHLOS Total N = 130880 |
|---|---|---|---|---|---|---|---|---|---|
| **Age (SD)** | 78.1 (6.3) | 62.0 (9.7) | 68.7 (6.4) | 60.8 (10.3) | 63.2 (7.1) | 61.6 (10.9) | 62.0 (9.4) | 63.8 (9.7) | 62.8 (10.1) |
| **Sex, n (%)** | | | | | | | | | |
| Female | 924 (49.9) | 7904 (54.5) | 1336 (53.1) | 18502 (56.1) | 1844 (49.9) | 5054 (56.6) | 7047 (55.1) | 29886 (55.9) | 72497 (55.5) |
| Male | 927 (50.1) | 6594 (45.5) | 1180 (46.9) | 14486 (43.9) | 1851 (50.1) | 3874 (43.4) | 5733 (44.9) | 23593 (44.1) | 58238 (44.5) |
| **Education, n (%)** | | | | | | | | | |
| Less than primary/ primary | 559 (33.0) | 4955 (36.6) | 1371 (54.5) | 8685 (26.3) | 1130 (30.7) | 4078 (45.7) | 9841 (78.7) | 13022 (24.7) | 43641 (34.0) |
| Secondary | 1018 (60.2) | 6346 (46.9) | 614 (24.4) | 18163 (55.1) | 2072 (56.3) | 3898 (43.7) | 1946 (15.6) | 28995 (55.1) | 63052 (49.1) |
| Tertiary | 115 (6.8) | 2227 (16.5) | 531 (21.1) | 6131 (18.6) | 477 (13.0) | 951 (10.6) | 711 (5.7) | 10637 (20.2) | 21780 (16.9) |
| **Wealth, n (%)** | | | | | | | | | |
| Quintile 1 (lowest) | 650 (36.2) | 2317 (17.7) | - | 6503 (19.7) | 509 (25.2) | 1821 (20.5) | 2954 (23.5) | 10119 (19.0) | 24873 (19.9) |
| Quintile 2 | 737 (41.0) | 2354 (18.0) | - | 6506 (19.7) | 378 (18.7) | 1796 (20.2) | 2388 (19.0) | 10369 (19.4) | 24528 (19.7) |
| Quintile 3 | 31 (1.7) | 2556 (19.5) | - | 6651 (20.2) | 438 (21.7) | 2235 (25.1) | 2467 (19.6) | 10555 (19.8) | 24933 (20.0) |
| Quintile 4 | 31 (1.7) | 2807 (21.4) | - | 6625 (20.1) | 314 (15.6) | 1428 (16.0) | 2370 (18.8) | 11129 (20.9) | 24704 (19.8) |
| Quintile 5 (highest) | 348 (19.4) | 3076 (23.4) | - | 6703 (20.3) | 379 (18.8) | 1616 (18.2) | 2397 (19.1) | 11164 (20.9) | 25683 (20.6) |
| **Smoking, n (%)** | | | | | | | | | |
| Ever smoked | 867 (50.3) | 8800 (61.2) | 1170 (46.5) | 19121 (58.3) | 1701 (46.5) | 2573 (28.8) | 5540 (43.0) | 24549 (46.6) | 64321 (49.6) |
| Never smoked | 856 (49.7) | 5581 (38.8) | 1346 (53.5) | 13696 (41.7) | 1960 (53.5) | 6355 (71.2) | 7350 (57.0) | 28163 (53.4) | 65307 (50.4) |
| **Drinking, n (%)** | | | | | | | | | |
| Often | 675 (36.9) | 4458 (31.6) | 1398 (55.6) | 6386 (19.4) | 1617 (44.6) | 1394 (36.8) | 783 (14.3) | 15191 (28.4) | 31902 (27.1) |
| Rare | 492 (26.9) | 8164 (57.8) | 221 (8.8) | 9513 (28.8) | 422 (11.7) | 2033 (53.6) | 1653 (30.1) | 21779 (40.7) | 44277 (37.6) |
| Never | 662 (36.2) | 1493 (10.6) | 894 (35.6) | 17087 (51.8) | 1584 (43.7) | 365 (9.6) | 3046 (55.6) | 16504 (30.9) | 41635 (35.3) |
| **Physical activity, n (%)** | | | | | | | | | |
| Sedentary/low | 1656 (90.6) | 3587 (26.5) | | 14106 (49.5) | 3142 (88.9) | 5788 (64.8) | - | 13946 (26.1) | 42225 (38.5) |
| Moderate | 143 (7.8) | 6749 (49.8) | | 10621 (37.3) | 343 (9.7) | 1050 (11.8) | - | 21916 (41.0) | 40822 (37.2) |
| High | 29 (1.6) | 3203 (23.7) | | 3759 (13.2) | 49 (1.4) | 2090 (23.4) | - | 17614 (32.9) | 36744 (24.3) |
| **Multimorbidity, n (%)** | | | | | | | | | |
| Presence | 708 (38.5) | 3816 (26.3) | 596 (23.7) | 9605 (29.1) | 560 (15.2) | 1490 (16.7) | 2855 (22.1) | 13188 (24.7) | 32818 (25.1) |
| Absence | 1132 (61.5) | 10682 (73.7) | 1920 (76.3) | 23383 (70.9) | 3127 (84.8) | 7438 (83.3) | 10070 (77.9) | 40290 (75.3) | 98042 (74.9) |

N = number, SD = standard deviation.

'rapid decline' or the 'low stable' groups. Compared to those who did not experience multiple illnesses simultaneously, people with multimorbidity were 1.7 times (OR = 1.7, 95% CI: 1.4–2.2) more likely to belong to the 'fast decline' group and 11.7 times (OR = 11.7, 95% CI: 10.9–12.6) more likely to belong to the 'low stable' group than the 'high stable' group (Table 3).

## Multimorbidity patterns and healthy ageing trajectory patterns in ELSA

At baseline, the average HAI score for the ELSA sample was 49.1 (SD: 9.4). The score decreased marginally over the follow up period: 48.8 (SD: 9.3) at Wave 3, 48.5 (SD: 9.2) at Wave 4, 48.4 (SD: 9.5) at Wave 5, 48.0 (SD: 9.5) at Wave 6 and 48.2 (SD: 9.6) at Wave 7. Since the multimorbidity status in ELSA was determined from 26 diseases, rather than eight as in ATHLOS, a considerably higher proportion of participants in ELSA (80.8%) reported to have had multimorbidity. ELSA sample characteristics from Wave 2 to Wave 7 are provided in S4 Table.

**Table 2. Model fit information–linear growth mixture model.**

| Number of classes | 2 classes | 3 classes | 4 classes | 5 classes |
|---|---|---|---|---|
| Sample size | 130880 | 130880 | 130880 | 130880 |
| Number of parameters | 19 | 22 | 25 | 28 |
| AIC | 3325789 | 3322654 | 3320917 | 3319442 |
| BIC | 3325974 | 3322869 | 3321161 | 3319716 |
| SABIC | 3325914 | 3322799 | 3321082 | 3319627 |
| LMR LR p-value | <0.001 | <0.001 | <0.001 | <0.001 |
| aLMR LR p-value | <0.001 | <0.001 | <0.001 | <0.001 |
| BLRT p-value | <0.001 | <0.001 | <0.001 | <0.001 |
| Entropy | 0.62 | 0.70 | 0.65 | 0.65 |
| Class size (%) | | | | |
| Class 1 | 76% | 76% | 52% | 55% |
| Class 2 | 24% | 2% | 1% | 2% |
| Class 3 | | 22% | 37% | 7% |
| Class 4 | | | 10% | 1% |
| Class 5 | | | | 35% |

AIC = Akaike information criteria, BIC = Bayesian information criteria, aBIC = adjusted Bayesian information criteria, LMR LR = Vuong-Lo-Mendell-Rubin likelihood ratio test, aLMR LR = adjusted Lo-Mendell-Rubin likelihood ratio test, BLRT = bootstrapped likelihood ratio test.

The GMM results for the baseline growth model using ELSA data were comparable to that using ATHLOS data. The three-class model was also found to be the optimal solution when ELSA data were used with an entropy statistic of 0.73, average posterior probability for each

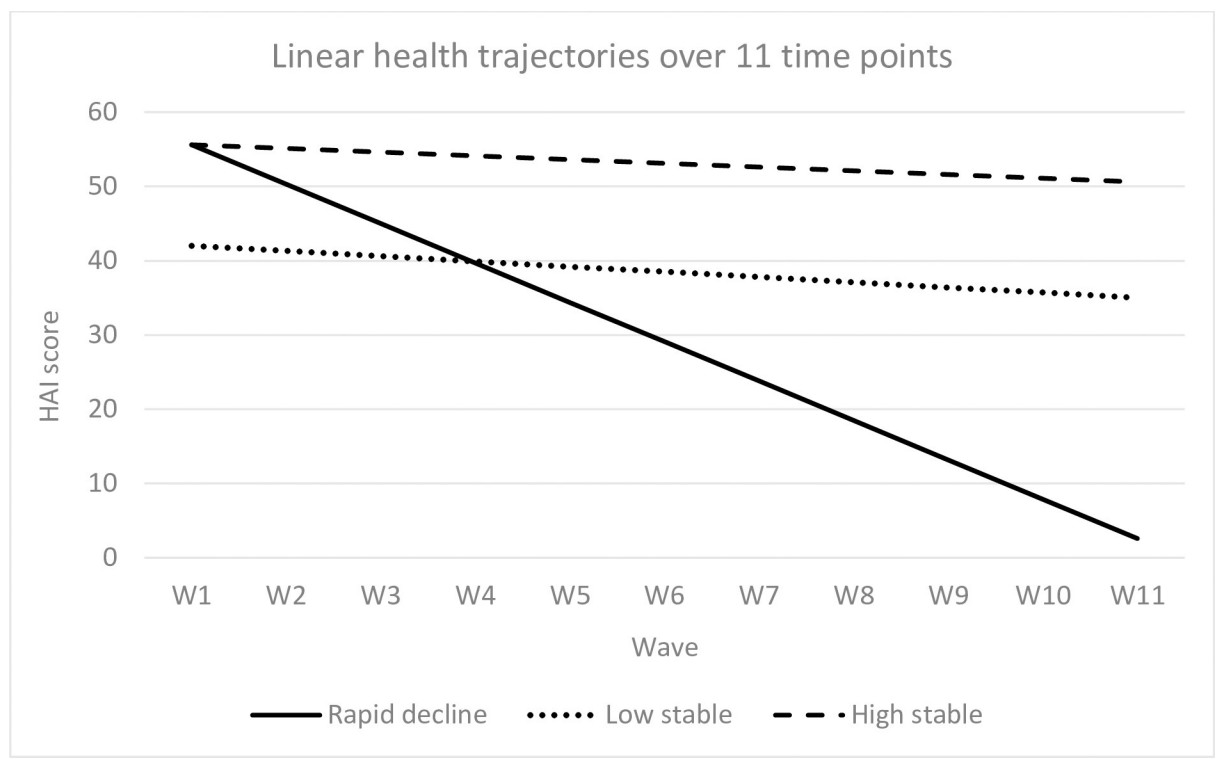

**Fig 1. Linear health trajectories over 11 time points.**

**Table 3. Multimorbidity status and healthy ageing trajectory patterns in ATHLOS.**

|  | High stable | Low stable | Rapid decline |
|---|---|---|---|
| N (%) | 100093 (76.5) | 28607 (21.9) | 2180 (1.7) |
| Mean intercept (SE) | 55.52 (0.05) | 41.96 (0.10) | 55.54 (0.29) |
| Mean slope (SE) | -0.55 (0.01) | -0.70 (0.02) | -5.34 (0.17) |
| Variance intercept (SE) | 27.81 (0.33) | | |
| Variance linear term (SE) | 0.82 (0.02) | | |
| Covariance intercept linear term (SE) | -1.01 (0.07) | | |
| **Multimorbidity status** | | | |
| Absence | Ref. | Ref. | |
| Presence (OR, 95% CI) | | 11.72 (10.92–12.57) | 1.71 (1.37–2.14) |

N = number, SE = standard error, OR = odds ratio, 95% CI = 95% confidence interval, Ref. = reference category.
Multinomial logistic regression was adjusted for age, sex, education, wealth, smoking, drinking and physical activity.

latent class above 0.70 and p-values of the likelihood ratio tests < 0.001 (S2 Table). Identical patterns of healthy ageing trajectories were identified [i.e. the 'high stable' group (61% of the sample), the 'low stable' group (36%) and the 'rapid decline' group (3%)] (Table 4).

When the baseline growth model was extended to include multimorbidity patterns and sociodemographic and lifestyle risk factors, the cardiorespiratory/arthritis/cataracts group was found to be associated with both the 'rapid decline' and the 'low stable' groups. Those who belonged to the cardiorespiratory/arthritis/cataracts group were 2.1 times (OR = 2.1, 95% CI: 1.2–3.8) more likely to have rapidly declining healthy ageing trajectories, compared to those in the relatively healthy group. These same individuals were, however, 9.8 times (OR = 9.8, 95% CI: 7.5–12.7) more likely to be assigned to the 'low stable' group, compared with the reference category. The multimorbidity pattern of metabolic diseases, on the other hand, was only associated with the 'low stable' group. People with this pattern of multimorbidity were three times (OR = 3.0, 95% CI: 2.2–4.0) more likely to have low HAI score at baseline, which declined gradually throughout the follow up period (Table 4).

## Discussion

This study described patterns of healthy ageing trajectories among 130880 individuals from eight longitudinal studies and examined their relationships with multimorbidity. We identified

**Table 4. Multimorbidity patterns and healthy ageing trajectory patterns in ELSA.**

|  | High stable | Low stable | Fast decline |
|---|---|---|---|
| N (%) | 5609 (61.2) | 3322 (36.2) | 239 (3.0) |
| Mean intercept (SE) | 54.56 (0.12) | 40.00 (0.17) | 51.97 (0.70) |
| Mean slope (SE) | -0.57 (0.04) | -0.55 (0.05) | -4.76 (0.39) |
| Variance intercept (SE) | 25.69 (0.88) | | |
| Variance linear term (SE) | 0.79 (0.06) | | |
| Covariance intercept linear term (SE) | -0.55 (0.21) | | |
| **Multimorbidity pattern** | | | |
| Relatively healthy | Ref. | Ref. | |
| Cardiorespiratory/arthritis/cataracts, OR (95% CI) | | 9.77 (7.50–12.73) | 2.09 (1.15–3.81) |
| Metabolic, OR (95% CI) | | 2.99 (2.23–4.00) | 1.08 (0.50–2.33) |

N = number, SE = standard error, OR = odds ratio, 95% CI = 95% confidence interval, Ref. = reference category.
Multinomial logistic regression was adjusted for age, sex, education, wealth, smoking, drinking and physical activity.

three healthy ageing trajectories patterns: 'high stable', 'low stable' and 'rapid decline'. We also found that people with multimorbidity, particularly those belonging to the cardiorespiratory/arthritis/cataracts group, were more likely to display worse healthy ageing trajectories than those without multimorbidity or relatively healthy.

Our results aligned with those reported in earlier studies [5,8], which identified similar patterns of healthy ageing trajectories in older adults in the ATHLOS cohorts. The existence of subgroups within the older population with distinctive projections of ageing trajectories of health indicated that healthy ageing is indeed a dynamic and heterogeneous process [44–46]. Our findings from the analysis of a subset of ATHLOS data also showed that there were identical patterns of healthy ageing trajectories among older adults in England, suggesting that the heterogeneity of healthy ageing was independent of the differences between countries. The majority of the people in both the ATHLOS and ELSA samples (98% and 97% respectively) were assigned to the 'high stable' and 'low stable' groups, showing that their trajectories of healthy ageing, represented by the HAI score, changed at a similar steady rate. Their HAI score at baseline, however, differed partly due to the existence of multimorbidity. Evidence from our study highlighted that although people with multimorbidity were more likely to have adverse health outcomes in general [47], those with more complex multimorbidity pattern, such as the cardiorespiratory/arthritis/cataracts group, were predicted to experience more dramatic changes in health [15,48,49]. The metabolic group, on the other hand, consisted of diseases that shared common pathological pathways, therefore may be more consistent in its trajectories of healthy ageing. Healthy ageing trajectories projected for different multimorbidity patterns may vary due to the severity and potential long-term impact of certain groups of diseases. Some health events such as myocardial infarction or stroke, even with timely treatments, are likely to impact more negatively on healthy ageing trajectories than, for example, the worsening of anaemia or iron deficiency.

The likelihood of belonging to the 'low stable' group was higher among people with multimorbidity, presumably because their level of healthy ageing at baseline was lower than those relatively healthy or without multimorbidity. The majority of our sample, nonetheless, were assigned to the 'high stable' group (77%), whose health at baseline was higher than average and only declined slightly over time. The same trend remained even when the analysis was repeated on a sample with 80% of participants with two or more chronic diseases (71% of whom were classified as being 'relatively healthy' despite the presence of multimorbidity [32]). This meant that even with multimorbidity, many people could still achieve satisfactory ageing if their health conditions were managed successfully. Once again, this supported the claim that individuals with multimorbidity were a heterogenous group, both in terms of the degree of complexity and types of diseases they experienced, and their healthy ageing capital (i.e. the preservation of intrinsic capacity and functional ability).

## Strengths and limitations

Our study was conducted on a large harmonised dataset, which included representative samples from 26 countries (i.e. Austria, Australia, Belgium, Czech Republic, Denmark, England, Estonia, France, Germany, Greece, Hungary, Ireland, Israel, Italy, Japan, Korea, Luxembourg, Mexico, the Netherlands, Poland, Portugal, Slovenia, Spain, Sweden, Switzerland and the US) in five continents. The development of a single health metric using a robust statistical method (IRT) was an innovative approach to conceptualising and measuring healthy ageing. The HAI score measured healthy aging as a latent construct and accounted for measurement errors, allowing for different patterns in participants' responses to observed items [4]. It differed from other measures that also attempted to examine the health status of older people, such as the

frailty phenotype [50] and frailty index [51], because it focused more on positive aspects of ageing and highlights the importance of the interaction between intrinsic capacity, functional ability and the environment, rather than simply summing up age-related biological and physiological deficits.

Our study was not without limitations, however. Although the ATHLOS harmonised dataset included data from eight cohorts, the representativeness of these cohorts varied at different waves. Only in the first three waves all eight cohorts were included. Nonetheless, results from our sensitivity analysis showed that the lack of representativeness from some cohorts thereafter did not affect the patterns of healthy ageing trajectories. The HAI score did not cover domains such as emotional stability, rendering the mental health dimension in older age underrepresented. Furthermore, since only common variables across studies were harmonised, the ATHLOS data may have lost some level of granularity in the original datasets. This was evident in the number of chronic diseases that was used to derived multimorbidity status [26 in ELSA vs. eight in ATHLOS]. The proportion of the sample with multimorbidity might be underestimated as a result. Furthermore, this reduced level of details on some diseases in the harmonised dataset hindered the use of ATHLOS data to account for the heterogeneous patterns of multimorbidity in eight cohorts. Although the healthy ageing trajectories were investigated longitudinally, we only considered multimorbidity status and patterns at baseline. Earlier research has shown that multimorbidity patterns can change over time [45,52,53]. For instance, from two health conditions at baseline an individual can get several more diseases throughout their lifetime and move from the 'relatively healthy' group to one of the more complex multimorbidity groups. Our study was based on self-reported data from the participants, thus prone to reporting biases. Since participants with a single observation or no information on the HAI score were excluded from the analytic sample, our study may be subject to selection bias. These participants might be at lower levels of health, which impeded them to participate or continue to participate in the studies. Their inclusion, if possible, could have led to the finding of an alternative, possibly worse healthy ageing trajectory. That being said, the baseline characteristics of the excluded sample were very similar to those of the analytic sample, as presented in S2 Table. This suggests that their exclusion might not have a big impact on the final results. Finally, our findings cannot be generalised to low- and middle-income countries (LMICs) due to their low presence in the dataset (of eight cohorts analysed in this study, only MHAS is a LMIC). Interpretations of our findings therefore must take these caveats into consideration.

## Conclusion

In this study we investigated the association between multimorbidity and different patterns of healthy ageing trajectories among 130880 individuals in a global sample. Multimorbidity appeared to increase the likelihood of having poorer healthy ageing trajectories, but the extent to which healthy ageing trajectories were projected to decline depended on the specific patterns of multimorbidity. With multimorbidity, it was still possible to achieve healthy ageing. Our findings reiterated that ageing is a heterogeneous process. Public health policies should therefore be implemented to account for this heterogeneity. Instead of regarding older people as frail and potential burdens of public health, policy makers and practitioners should actively promote healthy ageing in the recognition that ageing is not synonymous with ill health.

## Supporting information

**S1 Table. 41 items related to intrinsic capacity and functional ability used in the harmonisation process.**
(DOCX)

**S2 Table. Baseline characteristics of analytic and excluded samples.**
(DOCX)

**S3 Table. Model fit information–linear growth mixture model using the first three waves of the ATHLOS harmonised dataset.** Unadjusted GMM model using data from the first three waves of the ATHLOS harmonised dataset.
(DOCX)

**S4 Table. ELSA sample characteristics Waves 2–7.**
(DOCX)

**S5 Table. Model fit information–linear growth mixture model for ELSA dataset.**
(DOCX)

## Acknowledgments

The authors would like to thank Dr Albert Sanchez Niubo for his technical support and advice for this manuscript.

## Author Contributions

**Conceptualization:** Hai Nguyen, A. Matthew Prina.

**Formal analysis:** Hai Nguyen.

**Methodology:** Hai Nguyen, Dario Moreno-Agostino, Kia-Chong Chua, Silia Vitoratou, A. Matthew Prina.

**Supervision:** Kia-Chong Chua, Silia Vitoratou, A. Matthew Prina.

**Writing – original draft:** Hai Nguyen.

**Writing – review & editing:** Hai Nguyen, Dario Moreno-Agostino, Kia-Chong Chua, Silia Vitoratou, A. Matthew Prina.

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
