## [Decision Letter · Decision Letter 0]

6 Jan 2021

PONE-D-20-37601

Trajectories of healthy ageing among older adults with multimorbidity: a growth mixture model using harmonised data from eight ATHLOS cohorts

PLOS ONE

Dear Dr. Nguyen,

Thank you for submitting your manuscript to PLOS ONE. After careful consideration, we feel that it has merit but does not fully meet PLOS ONE’s publication criteria as it currently stands. Therefore, we invite you to submit a revised version of the manuscript that addresses the points raised during the review process.

We look forward to receiving your revised manuscript.

Kind regards,

Yiqiang Zhan

Academic Editor

PLOS ONE

Journal Requirements:

Reviewers' comments:

Reviewer's Responses to Questions

**Comments to the Author**

1. Is the manuscript technically sound, and do the data support the conclusions?

Reviewer #1: Partly

Reviewer #2: Yes

2. Has the statistical analysis been performed appropriately and rigorously? 

Reviewer #1: Yes

Reviewer #2: Yes

3. Have the authors made all data underlying the findings in their manuscript fully available?

Reviewer #1: No

Reviewer #2: Yes

4. Is the manuscript presented in an intelligible fashion and written in standard English?

Reviewer #1: Yes

Reviewer #2: Yes

5. Review Comments to the Author

Reviewer #1: The study investigated the associations between multi-morbidity and an aging index, called the healthy ageing index score, among a large study population. Data are originated from several independent cohorts across the world and cover a considerably large sample size. Yet the aims of this study are neither introduced nor answered properly. Major revisions are needed.

1. The authors used “healthy aging trajectory” to refer to “the longitudinal measures of healthy aging index (HAI) score” throughout the article. Yet I find “the healthy aging trajectory” an ambiguous concept, and I have no idea what it means exactly when reading the entire section of the introduction. I am particularly confused when reading through the aims of this study. I suggest to explain the concept of “healthy aging trajectory” at the very beginning of the article and explicitly use the word “HAI score” throughout.

2. The background of this study is largely missing and should be introduced. The authors spent two paragraphs to explain the importance of studying multi-morbidity and aging trajectory, yet failed to review the current evidence concerning the direct aims of this analysis: 1) HAI growth categories, 2) baseline multi-morbidity in association with HAI growth categories, 3) baseline multi-morbidity categories in association with HAI growth categories. I suggest the author review the literature more thoroughly and make a clear summary about “what is known” and “what is unknown.”

3. The HAI is the primary outcome of this study. It is an integrated score derived from 41 items. However, the methods of HAI development are poorly explained. Even though the authors referred to another independent article, some important details should be mentioned in the present manuscript. Such as how to assign value for each item? How to combine 41 values into a single score? What is the rational/statistical modeling of the combining algorithm? Is each item treated equally? Or does a different item have a different weight?

4. How are the HAI items measured? Self-reported?

5. How are the morbidities measured? Self-reported?

6. The first aim is “to describe different patterns of healthy ageing trajectories … globally”. In my opinion, a multi-ethnic study population is a strength of this study. However, the heterogeneity of different cohorts should be taken into account. It is not clear how the cohort effect is controlled in the entire analysis.

7. When it comes to trajectory patterns, there are at least two features to be considered, i.e., baseline level and changing rate. The authors identified three different patterns and 97% of the participants were classified as either “high stable” or “low stable” group. It seems like a majority of participants’ HAI changed at a similar rate, but their baseline HAI levels were different partly due to the existence of multi-morbidity. This is an interesting observation and could be discussed.

8. The authors considered HAI as “a more appropriate measure of healthy ageing than other indices (e.g. the frailty phenotype and frailty index).” Again, this is an ambiguous or even wrong claim. The author should be precise. In what aspects is HAI more appropriate than the frailty index?

9. The conclusion is too lengthy and, in my opinion, contains two many speculations and vague claims. For instance, what observation in the present analysis could lead to the last sentence of the conclusion part and what does “a new integrated care framework” means exactly? Inferences in the conclusion part should be based on your observations and be precise as well. Besides, I suggest avoiding references in the conclusion section.

Reviewer #2: The topic of the paper about healthy aging trajectory patterns and multimobility is interesting and impressive in aging field. The paper is well-written and the manuscript is logic and easy to understand. The author used the growth mixed model (GMM) to identify different trajectory patterns for healthy aging index (HAI) score and utilized the multinomial logistic regression to explore the association of multimorbidity status and patterns with healthy aging score trajectories patterns. The size of the full study sample including different longitudinal datasets in different countries is large to describe the patterns of HAI trajectories. And the methods and results could reach the aims of this study.

Minor comments:

1. The author need to clarify the trajectory pattern result in the abstract is from which dataset. Because this paper existed two HAI trajectory patterns for ATHLOS and ELSA data, respectively.

2. In the expression of the third aim: “the projection of healthy ageing trajectories over time”, “trajectory” itself own “over time” meaning in it. so “over time” might be a kind of repetition here. And “over time” used here might be a misleading that the study investigate the relationship between A and B with time.

3. One question about the pooled different national longitudinal datasets, is there any effect on the study result due to different ways of creating HAI or different calendar years in the datasets?

6. PLOS authors have the option to publish the peer review history of their article (what does this mean?). If published, this will include your full peer review and any attached files.

Reviewer #1: No

Reviewer #2: No

---

## [Author Response · Author response to Decision Letter 0]

15 Feb 2021

13 February 2021

Revise and re-submit: manuscript PONE-D-20-37601 “Trajectories of healthy ageing among older adults with multimorbidity: a growth mixture model using harmonised data from eight ATHLOS cohorts”

Dear Prof. Yiqiang Zhan,

We thank you for the opportunity to revise and resubmit our manuscript (PONE-D-20-37601) and for the perceptive and stimulating reviewers’ comments. We have studied the reviewers’ comments carefully and provided point-by-point responses to each of the comments in the following pages. 

We believe that the changes we have made have improved the paper and we hope that you will now find that it is suitable for publication in your journal. Once again, we would like to express our great appreciation to the reviewers for their comments on our manuscript. 

Sincerely yours,

Hai Nguyen 

RESPONSES TO REVIEWERS

Responses to reviewers 

Reviewer 1

Comment 1: The authors used “healthy aging trajectory” to refer to “the longitudinal measures of healthy aging index (HAI) score” throughout the article. Yet I find “the healthy aging trajectory” an ambiguous concept, and I have no idea what it means exactly when reading the entire section of the introduction. I am particularly confused when reading through the aims of this study. I suggest to explain the concept of “healthy aging trajectory” at the very beginning of the article and explicitly use the word “HAI score” throughout. 

Response 1: We are very grateful for the reviewer’s suggestion. We have added the following sentences to the introduction to clarify the concept of ‘healthy ageing trajectory’ (p.3, lines 67-73):

“To understand how people age and to respond to their healthcare needs, it is important to study how healthy ageing changes over time and what factors are associated with this process. A new measure of healthy ageing, the healthy ageing index (HAI) score, has been recently developed (Sanchez-Niubo et al 2020). As healthy ageing is a heterogeneous process, by analysing the growth patterns of the HAI score in the population, healthy ageing trajectories can be modelled.”

Sanchez-Niubo A, Forero CG, Wu YT, Gine-Vazquez I, Prina M, De La Fuente J, et al. Development of a common scale for measuring healthy ageing across the world: results from the ATHLOS consortium. Int J Epidemiol. 2020.

Comment 2: The background of this study is largely missing and should be introduced. The authors spent two paragraphs to explain the importance of studying multi-morbidity and aging trajectory, yet failed to review the current evidence concerning the direct aims of this analysis: 1) HAI growth categories, 2) baseline multi-morbidity in association with HAI growth categories, 3) baseline multi-morbidity categories in association with HAI growth categories. I suggest the author review the literature more thoroughly and make a clear summary about “what is known” and “what is unknown.” 

Response 2: We thank the reviewer for this comment. We have added the following sentences in the introduction to highlight the existing evidence about the relationship between multimorbidity and healthy ageing trajectory as well as the knowledge gap in this research area (p.4, lines 74-88). 

“Few studies have attempted to investigate healthy ageing trajectory patterns in older adults (Daskalopoulou et al 2019; Prina 2019; Wu et al 2020). Several patterns have been identified when examining the impact of lifestyle behaviours on healthy ageing (Daskalopoulou et al 2019; Moreno-Agostino et al 2020). However, there is little evidence about how these patterns might be different for people with multimorbidity. […] While multimorbidity (both its presence/absence and patterns) at baseline was found to be negatively associated with a baseline healthy ageing (Nguyen et al 2020), it is not clear how multimorbidity relates to the course of healthy ageing over time.”

Daskalopoulou C, Koukounari A, Wu YT, Terrera GM, Caballero FF, de la Fuente J, et al. Healthy ageing trajectories and lifestyle behaviour: the Mexican Health and Aging Study. Sci Rep. 2019;9(1):11041.

Prina AM. Health trajectories over time in the ATHLOS project: findings from multiple cohorts Innovation in Ageing. 2019;3(Supplement 1):S798.

Wu YT, Daskalopoulou C, Terrera GM, Niubo AS, Rodríguez-Artalejo F, Ayuso-Mateos JL, et al. Education and wealth quintiles in healthy ageing in eight harmonised cohorts in the ATHLOS Consortium: a population-based study. The Lancet Public Health. 2020;5(7):e386-e94.

Moreno-Agostino D, Daskalopoulou C, Wu YT, Koukounari A, Haro JM, Tyrovolas S, et al. The impact of physical activity on healthy ageing trajectories: evidence from eight cohort studies. Int J Behav Nutr Phys Act. 2020;17(1):92.

Nguyen H, Wu YT, Dregan A, Vitoratou S, Chua KC, Prina AM. Multimorbidity patterns, all-cause mortality and healthy aging in older English adults: Results from the English Longitudinal Study of Aging. Geriatr Gerontol Int. 2020.

Comment 3: The HAI is the primary outcome of this study. It is an integrated score derived from 41 items. However, the methods of HAI development are poorly explained. Even though the authors referred to another independent article, some important details should be mentioned in the present manuscript. Such as how to assign value for each item? How to combine 41 values into a single score? What is the rational/statistical modeling of the combining algorithm? Is each item treated equally? Or does a different item have a different weight? 

Response 3: We agree with the reviewer that the methods of HAI development should be elaborated. We have added a new reference to a recent study which describes in much detail how the HAI score was developed by the ATHLOS researcher (Sanchez-Niubo et al 2020). We have also added the following sentences to summarise important details of the construction of the HAI score in the methods section (p7, lines 144-163):

“IRT describes the relationship between an individual’s ‘trait’ (unobserved characteristic or attribute) and how they respond to items on a scale (Nguyen et al 2014). In this study, a two-parameter logistic IRT model was implemented. Specifically, two parameters were estimated for each item: difficulty (reflecting the level of the latent trait where the probability of endorsing a particular item was 0.50) and discrimination (reflecting the items ability to discriminate between different levels of the latent trait around the difficulty level). Thus, items contributed differently when estimating the respondents’ levels in the latent trait (Sanchez-Niubo et al 2020). […] The adequacy of the IRT model as a measurement scale was tested using the Root Mean Square Error of Approximation (RMSEA; good fit <0.06), the Comparative Fit Index (CFI; good fit >0.95) and the Tucker-Lewis index (TLI; good fit >0.95) (Sanchez-Niubo et al 2020). […] The score of each individual was calculated using the expected a posteriori estimation method (Sanchez-Niubo et al 2020).”

Nguyen TH, Han HR, Kim MT, Chan KS. An introduction to item response theory for patient-reported outcome measurement. Patient. 2014;7(1):23-35.

Sanchez-Niubo A, Forero CG, Wu YT, Gine-Vazquez I, Prina M, De La Fuente J, et al. Development of a common scale for measuring healthy ageing across the world: results from the ATHLOS consortium. Int J Epidemiol. 2020.

Comment 4: How are the HAI items measured? Self-reported? 

Response 4: The HAI score was constructed using both self-reported items and performance tests (e.g. immediate recall of common nouns from a list, verbal fluency). This is clarified in the manuscript as follows (p.7, lines 150-152):

“The two-parameter logistic IRT model estimated the latent trait score, which comprised 41 items (both self-reported and measured by performance tests), covering various domains of health and ageing (Supplement Table S1).”

It is also included as a footnote under Supplement Table S1:

N.B. “Items marked ꝉ were measured by performance tests (otherwise self-reported)”

Comment 5: How are the morbidities measured? Self-reported? 

Response 5: In the ATHLOS dataset, the diseases (morbidities) were indeed self-reported. We have added this information to the description of the independent variable ‘multimorbidity status’ (p.8, lines 176-180):

“Eight diseases were included in the ATHLOS dataset, namely, diabetes, asthma, chronic pulmonary disease, hypertension, arthritis, angina, myocardial infarction and stroke. These eight diseases were common across 16 studies included in the ATHLOS dataset, and was self-reported by the participants, following the question “Has a doctor ever told you that you have/have had any of the following disease?”.

In the ELSA dataset, however, the diseases were both self-reported and derived from objective measures during nurse visits (e.g. blood pressure, lung capacity). This was clarified in the description of the multimorbidity patterns at ELSA Wave 2 (p.8, lines 186-188):

“Multimorbidity patterns were used as a key predictor of healthy ageing trajectories patterns in the ELSA dataset. They were determined from 26 diseases at ELSA Wave 2 (both self-reported and objectively measured during nurse visits)”.

Comment 6: The first aim is “to describe different patterns of healthy ageing trajectories … globally”. In my opinion, a multi-ethnic study population is a strength of this study. However, the heterogeneity of different cohorts should be taken into account. It is not clear how the cohort effect is controlled in the entire analysis. 

Response 6: Thank you for this insightful comment. Due to the nature of the analysis (i.e. mixture model), we initially modelled the GMM using data from ATHLOS in an unconditional manner. Thus, the study variable (which represents cohorts) was not part of the class identification process. However, in step 3 of the 3-step approach (i.e. the multinomial model) this variable was included as a covariate to account for the potential source of confounding posed by different cohorts. We have clarified this in the methods section (p.11, lines 254-258), as follows:

“Finally, in step 3, a new model was estimated to evaluate the impact of predictor variables on the class membership, with measurement errors fixed to values obtained in step 2. The potential source of confounding posed by the heterogeneity of different cohorts included in the analysis was also accounted for by including the study variable in this step.”

Comment 7: When it comes to trajectory patterns, there are at least two features to be considered, i.e., baseline level and changing rate. The authors identified three different patterns and 97% of the participants were classified as either “high stable” or “low stable” group. It seems like a majority of participants’ HAI changed at a similar rate, but their baseline HAI levels were different partly due to the existence of multi-morbidity. This is an interesting observation and could be discussed. 

Response 7: We thank the reviewer for this helpful comment. This is indeed an important point that should be discussed. We have incorporated this into the Discussion section (p. 19, lines 373-377), as follows:

“The majority of the people in both the ATHLOS and ELSA samples (98% and 97% respectively) were assigned to the ‘high stable’ and ‘low stable’ groups, showing that their trajectories of healthy ageing, represented by the HAI score, changed at a similar steady rate. Their HAI score at baseline, however, differed partly due to the existence of multimorbidity”.

Comment 8: The authors considered HAI as “a more appropriate measure of healthy ageing than other indices (e.g. the frailty phenotype and frailty index).” Again, this is an ambiguous or even wrong claim. The author should be precise. In what aspects is HAI more appropriate than the frailty index? 

Response 8: We appreciate the reviewer’s comment. We have clarified how the HAI score differed from (rather than be more appropriate than) other indices such as the frailty phenotype and frailty index (p.21, lines 408-417), as follows:

“The development of a single health metric using a robust statistical method (IRT) was an innovative approach to conceptualising and measuring healthy ageing. The HAI score measured healthy aging as a latent construct and accounted for measurement errors, allowing for different patterns in participants’ responses to observed items (Sanchez-Niubo, Forero, Wu et al, 2020). It differed from other measures that also attempted to examine the health status of older people, such as the frailty phenotype (Fried, Tangen, Walston et al, 2001) and frailty index (Mitnitski, Mogilner & Rockwood, 2001), because it focused more on positive aspects of ageing and highlights the importance of the interaction between intrinsic capacity, functional ability and the environment, rather than simply summing up age-related biological and physiological deficits.”

Sanchez-Niubo A, Forero CG, Wu YT, Gine-Vazquez I, Prina M, De La Fuente J, et al. Development of a common scale for measuring healthy ageing across the world: results from the ATHLOS consortium. Int J Epidemiol. 2020.

Fried LP, Tangen CM, Walston J, Newman AB, Hirsch C, Gottdiener J, et al. Frailty in older adults: evidence for a phenotype. J Gerontol A Biol Sci Med Sci. 2001;56(3):M146-56.

Mitnitski AB, Mogilner AJ, Rockwood K. Accumulation of deficits as a proxy measure of aging. ScientificWorldJournal. 2001;1:323-36.

Comment 9: The conclusion is too lengthy and, in my opinion, contains two many speculations and vague claims. For instance, what observation in the present analysis could lead to the last sentence of the conclusion part and what does “a new integrated care framework” means exactly? Inferences in the conclusion part should be based on your observations and be precise as well. Besides, I suggest avoiding references in the conclusion section. 

Response 9: We are grateful to the reviewer for this helpful comment. Indeed, our conclusion included many new concepts that had not been introduced earlier in the manuscript. We have removed these as well as their associated references and made the conclusion more concise (pp.22-23, lines 451-460), as follows:

“In this study we investigated the association between multimorbidity and different patterns of healthy ageing trajectories among 130880 individuals in a global sample. Multimorbidity appeared to increase the likelihood of having poorer healthy ageing trajectories, but the extent to which healthy ageing trajectories were projected to decline depended on the specific patterns of multimorbidity. With multimorbidity, it was still possible to achieve healthy ageing. Our findings reiterated that ageing is a heterogeneous process. Public health policies should therefore be implemented to account for this heterogeneity. Instead of regarding older people as frail and potential burdens of public health, policy makers and practitioners should actively promote healthy ageing in the recognition that ageing is not synonymous with ill health”.

Reviewer 2

Comment 1: The authors need to clarify the trajectory pattern result in the abstract is from which dataset. Because this paper existed two HAI trajectory patterns for ATHLOS and ELSA data, respectively. 

Response 1: We thank the reviewer for this useful suggestion. We have now clarified the trajectory patterns and the datasets in the abstract (p.2, lines 40-43), as follows:

“Three similar patterns of healthy ageing trajectories were identified in the ATHLOS and ELSA datasets: 1) a ‘high stable’ group (76% in ATHLOS, 61% in ELSA), 2) a ‘low stable’ group (22% in ATHLOS, 36% in ELSA) and 3) a ‘rapid decline’ group (2% in ATHLOS, 3% in ELSA)”

Comment 2: In the expression of the third aim: “the projection of healthy ageing trajectories over time”, “trajectory” itself own “over time” meaning in it. so “over time” might be a kind of repetition here. And “over time” used here might be a misleading that the study investigate the relationship between A and B with time. 

Response 2: Thank you for pointing this out. We have now removed the phrase “over time” after ‘healthy ageing trajectories’ throughout the manuscript. 

Comment 3: One question about the pooled different national longitudinal datasets, is there any effect on the study result due to different ways of creating HAI or different calendar years in the datasets? 

Response 3: In longitudinal analyses, there is no way to de-confound the role of one of the age-period-cohort triad when two of them are included as chronological time. However, the effect of the starting time was controlled in part by the inclusion of the ‘study’ variable (which represents different cohorts in this study). As mentioned in our response 6 to Reviewer 1, due to the nature of our analysis (i.e. mixture model), we initially modelled the GMM using data from ATHLOS in an unconditional manner. Thus, the study variable was not part of the class identification process. However, in step 3 of the 3-step approach (i.e. the multinomial model) this variable was included as a covariate to account for the potential source of confounding posed by different cohorts (with different starting points). We have clarified this in the methods section (p.11, lines 254-258), as follows:

“Finally, in step 3, a new model was estimated to evaluate the impact of predictor variables on the class membership, with measurement errors fixed to values obtained in step 2. The potential source of confounding posed by the heterogeneity of different cohorts included in the analysis was also accounted for by including the study variable in this step.”

---

## [Decision Letter · Decision Letter 1]

8 Mar 2021

Trajectories of healthy ageing among older adults with multimorbidity: a growth mixture model using harmonised data from eight ATHLOS cohorts

PONE-D-20-37601R1

Dear Dr. Nguyen,

We’re pleased to inform you that your manuscript has been judged scientifically suitable for publication and will be formally accepted for publication once it meets all outstanding technical requirements.

Kind regards,

Y Zhan

Academic Editor

PLOS ONE

Additional Editor Comments (optional):

Reviewers' comments:

Reviewer's Responses to Questions

**Comments to the Author**

1. If the authors have adequately addressed your comments raised in a previous round of review and you feel that this manuscript is now acceptable for publication, you may indicate that here to bypass the “Comments to the Author” section, enter your conflict of interest statement in the “Confidential to Editor” section, and submit your "Accept" recommendation.

Reviewer #1: All comments have been addressed

Reviewer #2: All comments have been addressed

2. Is the manuscript technically sound, and do the data support the conclusions?

Reviewer #1: Yes

Reviewer #2: Yes

3. Has the statistical analysis been performed appropriately and rigorously? 

Reviewer #1: Yes

Reviewer #2: Yes

4. Have the authors made all data underlying the findings in their manuscript fully available?

Reviewer #1: No

Reviewer #2: Yes

5. Is the manuscript presented in an intelligible fashion and written in standard English?

Reviewer #1: Yes

Reviewer #2: Yes

6. Review Comments to the Author

Reviewer #1: The authors have responded to my questions properly.

I have no further comments.

Reviewer #2: Thanks for your revision based on my comments. You have answered all my comments but one place you need to remove "over time" in the objective part of the abstract (aim 3) and check this throughout your manuscript.

7. PLOS authors have the option to publish the peer review history of their article (what does this mean?). If published, this will include your full peer review and any attached files.

Reviewer #1: No

Reviewer #2: No

---

## [Editor Report · Acceptance letter]

29 Mar 2021

PONE-D-20-37601R1 

Trajectories of healthy ageing among older adults with multimorbidity: a growth mixture model using harmonised data from eight ATHLOS cohorts 

Dear Dr. Nguyen:

I'm pleased to inform you that your manuscript has been deemed suitable for publication in PLOS ONE. Congratulations! Your manuscript is now with our production department. 

Kind regards, 

on behalf of

Dr. Y Zhan 

Academic Editor

PLOS ONE